# α-Synuclein Conformational Plasticity: Physiologic States, Pathologic Strains, and Biotechnological Applications

**DOI:** 10.3390/biom12070994

**Published:** 2022-07-17

**Authors:** Amanda Li, Cyrus Rastegar, Xiaobo Mao

**Affiliations:** 1Neuroregeneration and Stem Cell Programs, Institute for Cell Engineering, Johns Hopkins University School of Medicine, Baltimore, MD 21205, USA; a.l.li@wustl.edu (A.L.); cyrus.rastegar@utsouthwestern.edu (C.R.); 2Washington University School of Medicine, Washington University in St. Louis, St. Louis, MO 63110, USA; 3University of Texas Southwestern Medical Center, Dallas, TX 75390, USA; 4Department of Neurology, Johns Hopkins University School of Medicine, Baltimore, MD 21205, USA

**Keywords:** α-synuclein, tetramer, α-synucleinopathy, prion-like, strains, PMCA, RT-QuIC

## Abstract

α-Synuclein (αS) is remarkable for both its extensive conformational plasticity and pathologic prion-like properties. Physiologically, αS may populate disordered monomeric, helically folded tetrameric, or membrane-bound oligomeric states. Pathologically, αS may assemble into toxic oligomers and subsequently fibrils, the prion-like transmission of which is implicated in a class of neurodegenerative disorders collectively termed α-synucleinopathies. Notably, αS does not adopt a single “amyloid fold”, but rather exists as structurally distinct amyloid-like conformations referred to as “strains”. The inoculation of animal models with different strains induces distinct pathologies, and emerging evidence suggests that the propagation of disease-specific strains underlies the differential pathologies observed in patients with different α-synucleinopathies. The characterization of αS strains has provided insight into the structural basis for the overlapping, yet distinct, symptoms of Parkinson’s disease, multiple system atrophy, and dementia with Lewy bodies. In this review, we first explore the physiological and pathological differences between conformational states of αS. We then discuss recent studies on the influence of micro-environmental factors on αS species formation, propagation, and the resultant pathological characteristics. Lastly, we review how an understanding of αS conformational properties has been translated to emerging strain amplification technologies, which have provided further insight into the role of specific strains in distinct α-synucleinopathies, and show promise for the early diagnosis of disease.

## 1. Introduction

Decades of work have illuminated the remarkable conformational plasticity of αS essential for its physiologic functions, but which also underlies its pathological properties in diseased states. In this mini-review, we seek to illustrate the unique conformational properties of αS and how they have been exploited by emerging technologies for the early diagnosis of disease. We begin by discussing the physiologic conformational states populated by αS and their functional properties (Section 2). We then discuss the pathologic process by which αS self-assembles into fibrils, the prion-like transmission of which underlies the pathophysiology of the synucleinopathies [1,2] (Section 3). Lastly, we discuss how a fundamental understanding of αS’ conformational properties have been translated to the development of strain amplification technologies, which show promise for the diagnosis of synucleinopathies, even before the onset of clinical symptoms [3] (Section 4).

## 2. Physiologic States

### 2.1. Monomer

#### 2.1.1. Structure

αS is a 140-amino acid protein consisting of an N-terminal amphipathic repeat region, central hydrophobic region, and C-terminal acidic region [4]. The intrinsically disordered nature of the soluble αS monomer is well established [5]. In contrast, αS adopts highly ordered α-helical conformations upon lipid binding. The biophysical properties of this interaction have since been extensively characterized [6]. Here, we will focus on recent advances in our understanding of the αS–lipid interaction and its biological functions.

In particular, recent reports have provided a structural basis for αS’ functions at membrane surfaces. These studies largely rely on acidic small unilamellar vesicles (SUVs) as a mimetic of synaptic vesicles [7]. Upon SUV binding, αS forms two α-helical “anchors” in the N-terminal (anchor #1) and central (anchor #2) portions of the protein, while the unstructured C-terminal region transiently interacts with the SUV’s surface [7,8]. The central anchor is highly sensitive to alterations in lipid composition, in contrast to the N/C terminal regions, which are largely invariant under such perturbations [7]. Thus, the central anchor additionally acts as a membrane “sensor,” conferring selectivity to the αS lipid-binding interaction (Figure 1). 

Interestingly, lipid binding by the N-terminal and central regions is decoupled, such that each region may not necessarily be associated with the same membrane. Specifically, the N-terminal anchor and central anchor of a single αS molecule may, at times, interact with different membranes (Figure 1) [9]. Simultaneous association with two synaptic vesicles promotes vesicle clustering [9], while association with a vesicle and the plasma membrane promotes tethering at membrane sites [10] (Figure 1). Membrane enrichment with gangliosides, seen in some neurodegenerative conditions, enhances αS’ interactions with the plasma membrane (PM), aberrantly affecting vesicle–PM dynamics [10].

#### 2.1.2. Function

Although these aforementioned reports have provided mechanistic insight into the lipid-bound αS state, they relied largely on artificial SUVs, which may not fully capture the dynamic and heterogeneous composition of membranes in vivo [11]. Indeed, the dynamic regulation of phosphatidylinositol phosphate (PIP) abundance may be an important determinant of αS membrane affinity. In cells, αS appears to be concentrated in foci rich in acidic PIPs, and the modulation of PIP acidity by PIP-kinases and phosphatases may regulate αS function in vivo [12]. 

Promisingly, a recently reported “cellular unroofing” approach has enabled the biophysical characterization of lipid-bound αS with native plasma membranes [13]. In this study, the authors “unroofed” the upper portion of the plasma membrane while leaving the lower portion intact. This strategy enabled the characterization of αS’ interactions with the inner leaflet of the plasma membrane (IPM). Interestingly, the conformational heterogeneity of IPM-bound αS was far greater than suggested by the SUV experiments [10]. Further studies with more physiologic systems will be crucial to determine how disease-related disturbances in lipid content contribute to the pathogenesis of synucleinopathies.

An additional finding from the “cellular unroofing” study was the localization of αS to exocytic sites, suggested by colocalization with SNARE proteins [13], which constitute the core fusion machinery for synaptic vesicle fusion and subsequent neurotransmitter release at the pre-synaptic membrane [14,15]. This is consistent with αS’ localization at the pre-synaptic membrane, where it functions as a chaperone to promote SNARE complex assembly [16]. Interestingly, chaperone activity appears to be contingent on the self-assembly of αS into large membrane-bound multimeric structures [17]. This multimeric conformation may constitute the functionally active form in the context of SNARE assembly [17], and appears to protect αS against aggregation into pathologic states [18]. 

### 2.2. Tetramer

The existence of a helically folded αS tetramer was first reported nearly a decade ago by two research groups using distinct approaches [19,20]. These initial studies suggested that, physiologically, αS predominantly populates an aggregation-resistant tetrameric state, and that the disassembly of the tetramer to an aggregation-prone monomer precedes disease. The pharmacologic stabilization of this putative tetrameric state could have therapeutic potential; however, its existence has proven highly controversial. Here, we first discuss evidence in favor of a tetrameric state, then the literature that failed to detect a physiologic αS tetramer.

#### 2.2.1. Evidence for an αS Tetramer

In 2011, Bartels et al. demonstrated, using multiple approaches, the possible existence of a tetrameric αS species. Initial evidence came from native polyacrylamide gel electrophoresis (Native-PAGE) of several cell lines, murine cortical samples, and human erythrocytes, which all contained a band corresponding approximately to the size of an αS tetramer. As Native-PAGE migration does not depend solely on molecular weight, the group developed an alternative in vitro crosslinking approach that is now frequently used to study the putative αS tetramer. In this technique, chemical crosslinking is performed prior to SDS-PAGE to preserve native assemblies. Lastly, the authors developed a method to purify natively folded αS directly from human erythrocytes. Both scanning transmission electron microscopy and analytical ultra centrifugation of purified αS supported the existence of a tetrameric species. Interestingly, tetrameric αS had a circular dichroism (CD) spectrum characteristic of an α-helical protein, was resistant to aggregation, and bound membrane phospholipids with substantially higher affinity than the monomer.

Shortly thereafter, a second group reported similar findings in bacterially expressed αS [20]. When purified under non-denaturing conditions, abundant αS tetramers and a smaller population of trimers were detected. NMR analysis suggested transient α-helix formation in two regions spanning residues 4–43 and 50–103, while the C-terminal region remained disordered. Interestingly, the αS tetramer underwent a cooperative, albeit irreversible, thermal unfolding transition, and thermally denatured αS readily formed fibrils, while native tetrameric αS did not. Taken together, these studies suggest that an α-helical, aggregation-resistant αS tetramer is a major physiologic species that constitutes the functional form of the protein [19,20]. 

Several reports have since delineated the basic properties of the αS tetramer, including the sequence features governing its formation. Intriguingly, tetramerization appears to be a general property of the synuclein protein family, which contains several imperfectly conserved KTKEGV repeat motifs throughout the protein sequence [21]. Although successive 10-residue deletion in αS failed to impair tetramer formation, the disruption of KTKEGV motifs by missense mutations reduced the population of the tetramer, as well as related minor multimeric species [21,22]. This suggests that tetramerization ability is not encoded locally, but is rather distributed throughout KTKEGV motifs interspersed within the αS sequence.

Furthermore, αS mutations implicated in familial PD (fPD) cause a shift in the populations of αS species in favor of the monomer in vitro and in vivo [23]. One such fPD mutation, E46K, occurs within a KTKEGV motif, and the “amplification” of this mutation in additional KTKEGV repeats causes the dose-dependent destabilization of the tetrameric state, inclusion formation, and neurotoxicity in vitro [21,23]. The insertion of three such E → K mutations was sufficient to cause PD-like motor dysfunction and pathology in a murine model (E3K), which could be partially rescued with the administration of L-DOPA [24]. 

These findings suggested that the stabilization of the tetrameric state may have therapeutic potential for PD, and led to the subsequent identification of cellular regulators of tetramer abundance [25,26,27]. In particular, GBA1 encodes β-glucocerebrosidase, a lysosomal enzyme that catalyzes the degradation of glucosylceramides [28]. GBA1 mutations lead to pathologic glucosylceramide accumulation and are a major risk factor for PD [29]. Interestingly, GBA1 knockout decreases the steady-state tetramer–monomer ratio, which can be rescued by the glucosylceramide synthase inhibitor miglustat [25]. This suggests that glucosylceramide concentration may modulate the steady-state abundance of tetrameric and monomeric αS.

Miglustat also appeared to be protective in an E3K model, suggesting the general importance of lipid content in the regulation of αS conformational states [26]. Indeed, more generally, membrane saturated fatty acids (SFAs) appear to stabilize the tetrameric state, while monounsaturated fatty acids (MUFAs) favor the monomeric state [26]. Inhibitors of stearoyl-CoA desaturase, which catalyzes the conversion of SFAs to MUFAs, are protective in vitro and in an E3K murine model of PD [27].

#### 2.2.2. Evidence against an αS Tetramer

The initial report of a physiologic tetramer [19] was controversial [30,31]. In particular, an early study by Burré et al. called into question Native-PAGE results identifying an apparent tetrameric species, suggesting that the increased apparent molecular weight may have resulted from the increased hydrodynamic radius of the unstructured monomeric protein [30]. We note, however, that this potential confounding factor was considered by the authors of the initial report [19], which motivated the development of an alternative in vitro crosslinking approach to support their results [19,22]. Additionally, Burré et al. failed to detect a tetrameric population in αS purified from mouse brain samples [30], which they suggested was more physiologically relevant than the erythrocyte-derived αS used in several experiments in the initial report [19]. Although the reasons for this discrepancy are unclear, subsequent studies later reported a 3:1 ratio of tetramer (and related multimeric species) to monomer in healthy human brain samples [23], and a tetrameric population was detected in induced pluripotent stem cell (iPSC)-derived human dopaminergic neurons [25].

Nevertheless, multiple experiments by Burré et al. support the absence of a soluble tetrameric αS conformation [30]. Specifically, brain-derived αS purified under non-denaturing conditions appears at an apparent tetrameric molecular weight by gel filtration, but retains this size after boiling [30]. Assuming the preservation of the putative tetrameric αS population during purification, this result is consistent with an increased apparent molecular weight due to an expanded, unstructured monomer [30], rather than a structured tetramer that irreversibly unfolds upon thermal denaturation [19,20]. Furthermore, size-exclusion chromatography coupled with multi-angle laser–light scattering (SEC–MALS) of freshly purified, brain-derived αS suggested the presence of only a major disordered monomeric population [30]. 

Additional studies have also failed to find evidence for a major tetrameric αS species [5,32,33,34,35]. In particular, in-cell NMR suggests that αS exists predominantly as an unstructured monomer in bacterial [32,35] and mammalian cells [5]. In the latter study, ^15^N-enriched αS was exogenously introduced into five mammalian cell lines, and approximately 90% of the introduced αS was detectable as unstructured monomers by NMR. This suggests that either (1) at most, a small population of αS could exist in a tetrameric state, in contrast to reports that three-quarters of αS are tetrameric in the healthy brain [23], or (2) exogenously introduced αS, despite experiencing the same cellular environment as native αS, cannot readily assemble into the putative tetramer, at least on the time scales of the experiment. 

## 3. Pathologic αS

In this section, we discuss the formation, structural properties, and mechanisms by which pathologic αS causes disease. We begin with a brief definition of “prion” and what is meant by the term “prion-like” often used in the literature to describe pathologic αS (Section 3.1). We then discuss the process by which physiologic αS assembles into the prion-like form, the αS amyloid fibril (Section 3.2). Next, we discuss the mechanisms by which pathologic αS spreads from cell to cell (Section 3.3). Lastly, we discuss evidence suggesting that there is not a single αS amyloid fold, but rather conformationally distinct “strains” of αS fibrils, the prion-like spread of which results in distinct patterns of pathology and clinical disease (Section 3.4).

### 3.1. Prion-like Properties of αS

The term prion was coined in 1982 by Stanley Prusiner in his description of “small proteinaceous infectious particles” as the infectious agent causing Scrapie [36,37]. Fundamentally, a prion is an infectious protein capable of perpetuating its structural information by inducing the conformational conversion of native proteins to the prion form [38]. In the literature, the pathologic form of αS is often referred to as a “prion” or “prion-like”, the latter referring to the lack of unequivocal evidence that prion spread of αS is a cause, rather than a consequence, of the disease process [39]. Nevertheless, there is substantial evidence supporting the notion that the spread of pathologic αS through a prion mechanism is causal in synucleinopathies [1,2,40,41,42,43,44,45,46] (for a discussion of this topic, see [47,48]). Early support for this hypothesis came from the observation that PD patients who had undergone experimental fetal ventral mesencephalic transplants subsequently developed Lewy body pathology in the grafted fetal neurons [41,42]. Perhaps the most compelling evidence for a prion mechanism was the demonstration by (1) Prusiner et al. that the inoculation of transgenic mice expressing human αS with brain homogenate from multiple system atrophy patients was sufficient to transmit disease, with αS pathology evident in postmortem analysis [43,44], and by (2) Virginia Lee et al. that a single inoculation of recombinant αS preformed fibrils (PFF) in vitro and in a non-transgenic mouse model led to the propagation of substantial αS pathology, which could be abolished by αS knockout in neurons and mice [1,45]. More recently, the transmission of αS has been demonstrated in vitro using pathologic αS derived from patients with Lewy body dementia [46]. In brief, emerging evidence has shown that pathogenic αS is a major driver in the pathogenesis of α-synucleinopathies.

### 3.2. Formation of the αS Fibril

The precise mechanism by which the αS monomer forms αS fibrils remains unclear; however, there is substantial evidence that this process requires the formation of conformationally heterogeneous pathologic αS oligomers [49,50,51,52]. αS oligomerization is driven by lipid interactions resulting in the formation of oligomers with variable β-sheet structure [49,52]. These pathologic αS oligomers are inherently neurotoxic, with the etiology of toxicity likely multifactorial, involving electrolyte disturbances and reactive oxygen species formation (ROS) secondary to the disruption of membrane integrity [52,53,54,55]. Although oligomerization precedes the formation of amyloid fibrils, these two processes are not necessarily directly coupled, as the C-terminal truncation of αS increases the rate of fibril formation, but reduces the rate of formation of toxic oligomers [56]. Physiologically, cellular mechanisms exist to disassemble pathologic αS conformations, including an Hsc70-based system that preferentially disassembles pathologic oligomers and short αS fibrils, with little activity against larger αS amyloid fibrils [57].

### 3.3. Propagation of the αS Fibril

In vivo, αS fibrils may spread from cell to cell in a prion-like manner. The origin of αS spread in vivo has not been definitively established; however, there is substantial evidence that αS may spread in a retrograde manner via the gut–brain axis, supported by (1) the epidemiologic association of gastrointestinal dysfunction with PD, (2) experimental models demonstrating the gut-to-brain spread of αS fibrils, and (3) evidence of the cross-fibrillization of amyloidogenic microbial proteins with αS (reviewed in [58]).

The spread of pathologic αS requires cellular uptake, the induction of aggregation of a physiologic population of αS, and the release of newly generated pathologic αS. Multiple cellular receptors have been described for αS fibril spread, including heparan sulfate proteoglycans, TLR2, neurexins, Na^+^/K^+^-ATPase subunit α3, Lag3, Aplp1, FcγRIIb, and the cellular prion protein [2,59,60,61,62,63,64,65,66,67,68,69]. The transmission of αS oligomers via an exosomal route has also been described, although there are conflicting reports regarding whether this exosomal pathway contributes to disease progression or is protective by reducing the intracellular pathologic αS burden when the capacity of the degradation machinery has been exceeded [70,71,72,73]. Regardless, receptor-mediated αS fibril transmission appears to play a vital role in the disease process. This is suggested by evidence that two neuronally expressed αS fibril receptors, Lag3 and Aplp1, form a complex crucial for αS uptake and the knockout of either receptor substantially reduces pathologic αS spread, with double knockout largely sparing dopaminergic neurons in vivo [2,67,68,69]. Mechanistically, receptor binding is driven by electrostatic interactions between the acidic C-terminal region of αS fibrils and basic regions of the Lag3 D1 and Aplp1 E1 domains [67]. These findings raise the exciting possibility that the modulation of αS receptor expression or pharmacologic receptor blockade may have therapeutic potential in the treatment of synucleinopathies.

An important consideration when discussing the transmission of αS is that the vast majority of pathologic αS has undergone post-translational modifications (PTMs), including phosphorylation, ubiquitination, nitration, truncation, and other modifications [74,75]. While a thorough review of αS PTMs has been described elsewhere [74] and is beyond the scope of this review, here, we will briefly discuss key PTMs and their structural and functional implications. In particular, the most abundant population of αS in Lewy bodies is mono-o or di-ubiquitinated and phosphorylated [76]. The SUMOylation of αS has also been described; however, it does not appear to be specific to the synucleinopathies, as increased SUMOylated αS has also been reported in Huntington’s disease and amyotrophic lateral sclerosis [74]. The accumulation of both ubiquitinated and SUMOylated αS likely reflects a failure of cellular proteostasis [74,77]. Phosphorylation sites appear concentrated in the C-terminal portion of the protein, with the most prevalent phosphorylation event occurring at Ser-129 [78,79]. Importantly, Ser-129 phosphorylation is infrequent in healthy individuals (~4% of the total αS population), but constitutes the vast majority (>90%) of αS in patients with PD [74]. Although it remains unclear whether this PTM contributes to or is protective against disease [75], there is some evidence that Ser-129-phosphorylated αS is a more potent inducer of reactive oxygen species than its unmodified counterpart [80], and may also enhance the transmission of αS consistent with the electrostatically driven nature of the αS fibril–receptor binding interaction [67]. Therefore, the most common PTM in pathologic αS, Ser-129 phosphorylation, appears to be both inherently more toxic and more transmissible than unphosphorylated αS.

The mechanism by which αS fibril propagation leads to neuronal death, either directly or indirectly, remains unclear. As discussed previously, there is substantial evidence that the pathologic oligomers formed by αS are inherently neurotoxic, in part through ROS formation [52]. Importantly, a recent study demonstrated that ROS scavenging with an antioxidant nanozyme is protective against pathologic αS-induced neurodegeneration in neuron culture and in a mouse model [81]. Additionally, in vivo, pathologic αS fibrils are found in lipid-rich inclusions termed Lewy bodies. The sources of these lipids include membrane fragments, mitochondria, and vesicular structures, including lysosomes and autophagosomes [82]. There is some evidence that the process of Lewy body formation, rather than the αS fibrils themselves, are neurotoxic, in part by the induction of mitochondrial damage and synaptic dysfunction [82,83]. Interestingly, a recent study demonstrated that the flavonoid dihydromyricetin, despite promoting αS fibrillization, is neuroprotective in vitro [84]. This protective effect appears to be due to the formation of a non-pathogenic αS fibrillar structure, or “strain” [84]. We will discuss this concept of αS strains in the subsequent section.

### 3.4. Cellular Determinants of Strain Formation

The synucleinopathies may broadly be classified as Lewy body diseases (LBDs), characterized by neuronal αS-rich inclusions (Lewy bodies), or multiple system atrophy (MSA), characterized predominantly by glial cytoplasmic inclusions (GCIs) in oligodendrocytes. Interestingly, brain-derived αS purified from Lewy bodies (LB-αS) and GCIs (GCI-αS) differ in their phosphorylation status, protease accessibility, and recognition by conformationally sensitive αS antibodies [85]. Together, these results suggest that LB-αS and GCI-αS populate conformationally distinct αS strains. Pathologically, GCI-αS is a more potent inducer of αS aggregation than LB-αS, consistent with the more aggressive nature of MSA [44,85].

In a landmark study, Peng et al. initially hypothesized that cell tropism may be conformationally encoded in the GCI-αS and LB-αS strains, giving rise to the selective involvement of oligodendrocytes and neurons in MSA and LBDs, respectively (2018). According to this hypothesis, one would expect GCI-αS to preferentially affect oligodendrocytes and LB-αS to preferentially affect neurons. Indeed, in primary oligodendrocytes, GCI-αS is ~1000× more potent at seeding aggregation than LB-αS. Contrary to this hypothesis, however, GCI-αS retains this 1000-fold potency over LB-αS in primary neurons. Furthermore, the inoculation of wild-type mice with GCI-αS produces more potent pathology than LB-αS, but in neurons rather than oligodendrocytes. Thus GCI-αS and LB-αS strains, despite their respective localizations to oligodendrocytes in MSA and neurons in LBD, lack any intrinsic specificity for these cell types.

Given that the strain type does not appear to determine cell-type involvement, the authors propose the converse hypothesis: cell-type involvement (i.e., oligodendrocyte or neuron-specific factors) determines the strain type formed (GCI-αS or LB-αS). Under this hypothesis, the exposure of an oligodendrocyte with GCI-αS or LB-αS should seed the formation of a GCI-αS strain. Indeed, in transgenic mice expressing αS only in oligodendrocytes, inoculation with either GCI-αS or LB-αS induces the formation of a GCI-αS strain. Furthermore, incubation with oligodendrocyte lysate was sufficient to generate GCI-αS [85]. Taken together, this suggests that oligodendrocytes contain specific cellular factors capable of strain conversion (LB-αS to GCI-αS).

Interestingly, although oligodendrocytes can convert LB-αS to GCI-αS, neurons appear incapable of the reverse process (GCI-αS to LB-αS). Given this irreversible conversion, why then do LBD patients lack the GCI-αS strain (i.e., oligodendrocyte inclusions)? Importantly, αS is not typically expressed in oligodendrocytes, but is either overexpressed or extracellularly acquired in MSA [86,87]. Therefore, in the absence of αS-containing oligodendrocytes (i.e., LBD patients), the LB-αS strain will persist, resulting in predominantly neuronal LB pathology. However, in MSA patients, the oligodendrocyte expression of αS enables the irreversible conversion of LB-αS to GCI-αS, resulting in predominantly oligodendrocyte GCI pathology (Figure 2) [85]. Taken together, this study provides compelling evidence that components of the cellular environment can directly influence strain formation. The identification of the specific oligodendrocyte factors responsible for strain conversion and the circumstances in which oligodendrocytes express αS will provide important insight into the pathophysiology of the synucleinopathies.

## 4. Applications of Strain-Amplification Techniques

Protein misfolding cyclic amplification (PMCA) has emerged as a powerful tool to study the αS self-aggregation process, characterize the properties of different αS strains, and, more recently, as a sensitive and specific method to diagnose and distinguish distinct α-synucleinopathies. PMCA was first described in 2001 by Soto et al. as a method conceptually similar to polymerase chain reaction cycling, but with the purpose of amplifying minute amounts of a misfolded protein above biochemical limits of detection [88]. Generally, minute quantities of prion protein template units are fragmented by sonication to make polymerization points available for the misfolding of wild-type proteins. This process is cyclically repeated to amplify misfolded proteins to detectable levels, and analyzed with proteinase K digestion and Western blot analysis for the identification of pathological aggregates [88]. Similar in concept, real-time quaking-induced conversion (RT-QuIC) replaces sonication with vigorous intermittent shaking to promote seeded aggregation, and replaces Western blot analysis with the real-time monitoring of fluorescence emitted by aggregate-sensitive Thioflavin-T dye (ThT) during the aggregation process [89]. An overview of strain amplification techniques with αS is shown in Figure 3.

Synucleinopathies are currently diagnosed based on clinical criteria, with definitive diagnosis only available post-mortem. However, here we explore recent advances in strain amplification techniques that show promise for the early, objective diagnosis of synucleinopathies [90].

### 4.1. αS Detection through Strain Amplification Assays

The detection of the prion-like form of αS with strain amplification assays, PMCA and RT-QuIC, may have both prognostic and diagnostic value. Here, we examine examples of each in the detection of αS in CSF and brain samples of patients with synucleinopathies. Both PMCA and RT-QuIC were modified for αS detection through fluorescence detection (ThT) instead of Western blot analysis, but differ in protocol, including pH, shaking conditions, and source of recombinant αS protein [91].

#### 4.1.1. Detection of αS in CSF and Brain

In 2017, the Soto group adapted PMCA to detect αS in the CSF of participants with and without PD, which was the first instance of the use of αS-PMCA as a biochemical diagnosis tool [3]. Their PMCA technique was sensitive and specific in detecting αS in subjects with PD versus controls (Table 1). There were also notable kinetic parameters of the PMCA reaction that correlated with disease severity at the time of sample collection. This study demonstrated the potential applications of PMCA in diagnosis and in monitoring the progression of synucleinopathies [3]. The following year, Becker et al. also used PMCA to detect the seeding activity of αS in formaldehyde-fixed MSA samples, and showed that PMCA with sonication was a sensitive and quantitative method for detecting αS seeding activity [92].

In 2016, RT-QuIC was applied by the Green group to detect αS in the CSF and brains of subjects with dementia with LB and PD compared with controls (Table 1). This illustrated the feasibility of RT-QuIC in the early clinical assessment of patients with synucleinopathies [93]. Further exemplifying how the strain-amplification of αS can be an early biomarker for synucleinopathies, Iranzo et al. used RT-QuIC in a longitudinal study following patients with isolated rapid-eye-movement sleep behavior disorder (IRBD) [94]. IRBD has been identified as a potential prodromal stage of the synucleinopathies PD and dementia with LB. CSF from patients with IRBD and controls was collected and analyzed with RT-QuIC (Table 1). Kaplan–Meier analysis showed that participants who were αS-negative had a lower risk of developing Parkinson’s disease or dementia with Lewy bodies than participants who were αS-positive. This is an exciting development demonstrating the potential of RT-QuIC to diagnose PD or DLB, even before the onset of clinical symptoms. Early detection may facilitate neuroprotective interventions prior to extensive neuronal injury [94].

#### 4.1.2. Detecting αS with Seeding Aggregation Assays in Non-CSF Samples

Most seeding aggregation assay studies for αS have relied on CSF samples, but pathologic αS has been measured in other peripheral tissue and bodily fluids. In PD, increased αS measurements have been collected in post-mortem and ante-mortem peripheral tissue samples of the cardiac plexus, sympathetic ganglia, gastric myenteric plexus, colonic tissue, GI tract, cardiac sympathetic nervous system, heart, salivary gland, and vagus nerve, with increasing evidence that the skin may also be a possible site of αS detection with conventional techniques, such as ELISA and multiplex immunoassays. While CSF has been the most reliably studied bodily fluid for αS detection, αS has also been detected in plasma or serum, blood, and saliva, with red blood cells as a major source of αS [95,96]. In addition to CSF, PMCA and RT-QuIC have been examined with other sites, including peripheral tissue obtained through routine GI and skin biopsies. These additional sites can be more easily accessed or more commonly obtained than CSF, enabling routine screening and the early detection of synucleinopathies.

αS-PMCA was applied to routine GI biopsies by Fenyi et al. Ten out of the eighteen PD patients had detectable αS aggregates, while only one control of eleven was αS-PMCA-positive. Interestingly, although this control was asymptomatic at the time of biopsy, they developed PD symptoms at a 10-year follow-up [97]. This application of seeding aggregation assays in GI biopsies may not be very sensitive, as only 10 out of 18 PD patients were αS-PMCA-positive, but it may be a specific test with utility for screening applications.

In addition to GI biopsies, skin αS seeding through seeding aggregation assays are another potential biomarker for synucleinopathies. The seeding activity of skin αS was analyzed with RT-QuIC and PMCA assays on 160 autopsies and 41 biopsies, and differentiated the PD samples from controls (Table 1) [98]. With these sensitive and specific techniques to assess skin αS seeding activity, skin samples may be used in the antemortem diagnosis of PD and other synucleinopathies. However, studies on other non-CSF or brain samples have been limited, and there is potential for additional investigation in easily accessed bodily fluids, especially blood, as a known major source of αS. Further investigation in additional peripheral tissue sites or bodily fluids will provide insight into the feasibility of PMCA and RT-QuIC for the minimally invasive routine diagnosis of synucleinopathies.

#### 4.1.3. Strain Amplification Assays in Differentiating Synucleinopathies

Prior studies focused mainly on using seeding aggregation assays to distinguish synucleinopathies from non-synucleinopathies, but recent developments in our understanding of kinetics, fibrillary aggregate structures, and advancements in technology have improved the discriminating ability of these assays to distinguish between synucleinopathies.

Shahnawa et al. of the Soto group were the first to establish an αS-PMCA assay to discriminate between samples of CSF from patients diagnosed with PD or MSA with high sensitivity and specificity. PMCA analysis indicated differences in the maximal ThT fluorescence and aggregation kinetics between CSF samples from PD versus MSA patients, which was confirmed by post-mortem brain specimens of the same subjects. Additional spectroscopic, biochemical, and toxicity studies also showed differences in structure and toxicity between MSA and PD αS aggregates [99].

The assessment of PMCA aggregation kinetics was conducted by the Soto group, as measured by ThT fluorescence. MSA samples aggregated faster, but reached a lower maximum fluorescence than PD samples. The group confirmed with post-mortem brain specimens of both PD and MSA patients that the activity of aggregates in CSF reflects that of aggregates in the brain. PMCA analysis distinguished PD from MSA, as well as PD and MSA from non-synucleinopathy controls (Table 1). After PMCA amplification, both MSA and PD samples contained equal amounts of aggregates, indicating that the difference in the maximal ThT fluorescence resulted from differences in ThT–aggregate interactions due to conformationally distinct αS prions in PD and MSAs. In examining structural differences, spectroscopy revealed that MSA aggregates have a higher proportion of β-sheet structures than PD aggregates. Additional studies of the aggregates with cryo-electron microscopy showed that both PD and MSA fibrils were composed of two protofilaments that intertwine in a left-handed helix. However, the structures of αS aggregates differed as MSA αS filaments had shorter twists (46–105 nm in length), whereas PD αS filaments were straighter, with longer helical twists (76.6–199 nm in length). This is consistent with immune-electron microscopy that showed brain-derived αS filaments from patients with MSA as predominantly twisted compared with the mostly straight filaments from PD patients. Additionally, in a cell culture model, MSA-derived αS-PMCA aggregates showed significant toxicity at concentrations four times lower than those of their PD-derived counterparts [99,100].

**Table 1 biomolecules-12-00994-t001:** **Sensitivities and Specificities of Strain-Amplification Assays**. The sensitivities and specificities from the studies described are listed in order of discussion, and categorized by sample source, application, and strain amplification technique.

Sample Source	Application	Technique	Sensitivity	Specificity	Reference
CSF	Differentiation of PD from non-synucleinopathy controls	PMCA	88.5%	96.6%	Shahnawaz et al., 2017 [3]
CSF	Differentiation of DLB from controls	RT-QuIC	92%	95%	Fairfoul et al., 2016 [93]
	Differentiation of PD from controls		95%	95%	
CSF	Differentiation of PD from controls	PMCA	95.2%	89.9%	Kang et al., 2019 [91]
		RT-QuIC	96.2%	82.3%	
		PMCA and RT-QuIC	97.1%	92.5%	
CSF	Differentiation of IRBD participants from controls	RT-QuIC	90%	90%	Iranzo et al., 2021 [94]
GI biopsies	Differentiation of PD from controls	PMCA	55.56%	81.81%	Fenyi et al., 2019 [97]
Autopsy skin biopsies	Differentiation of PD from controls	PMCA	82%	96%	Wang et al., 2020 [98]
		RT-QuIC	94%	98%	
Living skin biopsies		PMCA	80%	90%	
		RT-QuIC	95%	95%	
CSF	Differentiation of MSA from PD	PMCA	95.4%	100%	Shahnawaz et al., 2020 [99]
	Differentiation of PD from non-synucleinopathy controls		93.6%	100%	
	Differentiation of MSA from controls		84.6%	100%	
CSF	Differentiation of MSA from PD/DLB (maximum ThT fluorescence cutoff of <2000 AU)	PMCA	100%	83%	Singer et al., 2020 [101]
	Differentiation of MSA from controls (maximum ThT fluorescence cutoff of >150 AU)		97%	100%	

Additionally, in 2020, Singer et al. of the Low group, in collaboration with Soto, used PMCA to differentiate MSA from other Lewy body synucleinopathies in CSF samples [101]. The group observed the same pattern of αS aggregation kinetics as the Soto group in the previously discussed section, where PD/DLB aggregation occurred later with a higher maximum fluorescence level, which allowed them to differentiate MSA from PD/DLB. The group adjusted the cutoff for maximum ThT fluorescence to optimize the sensitivity and specificity of the assays to differentiate MSA from controls and MSA from PD/DLB (Table 1). The sensitivity and specificity are improved when ThT fluorescence is coupled with a neurofilament light-chain protein (NFL) biomarker cutoff, which is significantly elevated in MSA patients compared with healthy controls. The group found a range of cutoff values for both NFL and fluorescence values to differentiate MSA from PD/DLB and controls with high sensitivity and specificity [101].

Overall, PMCA was shown to maintain the biochemical and structural properties of disease-specific αS strains during the amplification process, and thus may serve as an important tool in distinguishing between MSA and PD, which may present with similar early clinical signs, despite their distinct progressions and treatment plans [98]. Without any current objective ante-mortem diagnosis tool for MSA and other synucleinopathies, PMCA is a potential non-invasive tool to analyze αS activity as a potential prognostic biomarker.

### 4.2. Comparing PMCA and RT-QuIC Detection of αS

A comparison of αS-PMCA and RT-QuIC was performed by Kang et al., who provided the Soto and Green groups with separate aliquots of CSF from the same subjects. Between the two techniques, there was a high concordance of results of 92%. Among the discrepant findings, no systematic explanation was found [91].

PMCA and RT-QuIC differ slightly in their protocol conditions, but both techniques apply a maximum ThT fluorescence threshold to identify a positive case of disease. The ThT dye is specific to aggregated fibrils, and is used in real-time to measure the kinetics of αS aggregation. Cutoff values for maximum ThT fluorescence are chosen to differentiate a sample with synucleinopathy from controls. Both assays were sensitive and specific, but the sensitivity and specificity were improved when both assays were performed together for concordant results (Table 1). In principle, both techniques support a common conceptual basis that misfolded αS proteins present in the CSF of PD patients can seed the aggregation of monomeric αS protein through a prion-like propagation of an aberrant protein conformation [1].

In summary, the self-aggregation assays, PMCA and RT-QuIC, show promise for the screening and diagnosis of Parkinson’s disease and other α-synucleinopathies. Recent advances in these techniques have enabled the identification of the conformationally distinct αS strains that underlie the pathophysiology of different α-synucleinopathies. They are objective tools with the capacity for early diagnosis, raising the possibility for enhanced patient care through early interventions targeting pathologic αS transmission or other neuroprotective measures. An enhanced understanding of the kinetics of αS aggregation, structural properties of αS fibrils, and coupling to additional biomarkers may further enhance the accuracy of these assays in the diagnosis and prognosis of synucleinopathies.

## 5. Conclusions

The αS protein is remarkable for its ability to populate a diverse set of conformational states—both physiologic and pathologic. Physiologically, the αS monomer is largely disordered in the cytosol, but adopts a highly ordered α-helical conformation upon membrane binding. Membrane-bound αS plays important physiological roles in synaptic vesicle clustering, fusion, and tethering at plasma membrane surfaces (Figure 1). In addition to these aforementioned states, αS may exist as a cytosolic, aggregation-resistant tetramer in vivo. This controversial tetrameric state remains an area of further investigation, as multiple groups using distinct approaches have both confirmed or refuted the existence of a major tetrameric population. In diseased states, pathologic αS forms conformationally diverse oligomeric species and amyloid fibrils. The oligomerization of αS may be driven by lipid interactions, and αS oligomers are inherently neurotoxic due to the disruption of membrane integrity. αS oligomers may form conformationally distinct amyloid fibrils, termed strains, with prion-like properties. The cell-to-cell spread of distinct strains may contribute to the progression of neurodegeneration with distinct patterns of αS pathology and clinical manifestations that constitute the synucleinopathies (Figure 2). Numerous receptors have been described for the transmission of pathologic αS, with further investigation warranted to pursue the therapeutic potential of the pharmacologic blockade or downregulation of these receptors in the treatment of synucleinopathies.

Strain amplification assays, PMCA and RT-QuIC, are powerful emerging techniques that can detect misfolded αS to diagnose and differentiate synucleinopathies. The prion-like seeding activity of αS in CSF, skin, and GI biopsy samples is a highly sensitive and specific biomarker for the presence of a synucleinopathy (Table 1). The utility of these assays is underscored by their ability to provide an objective, early diagnosis of disease, even before the onset of clinical symptoms, as well as their prognostic value (Figure 3). Further investigation should be performed with additional easily accessed peripheral tissue sites or bodily fluids, such as saliva, plasma, serum, and blood, which has abundant αS. Notably, PMCA has been used to differentiate MSA from PD and DLB based on differences in seeding kinetics. These differences in seeding kinetics have also been tied to differences in the conformational properties of disease-specific αS strains. However, further work needs to be conducted on the strain amplification kinetic criteria to differentiate synucleinopathies. With improved understanding of αS aggregation, strain amplification assays have important implications in clinical screening for disease, making differential diagnoses, and following the progression and prognosis of disease.

## Figures and Tables

**Figure 1 biomolecules-12-00994-f001:**
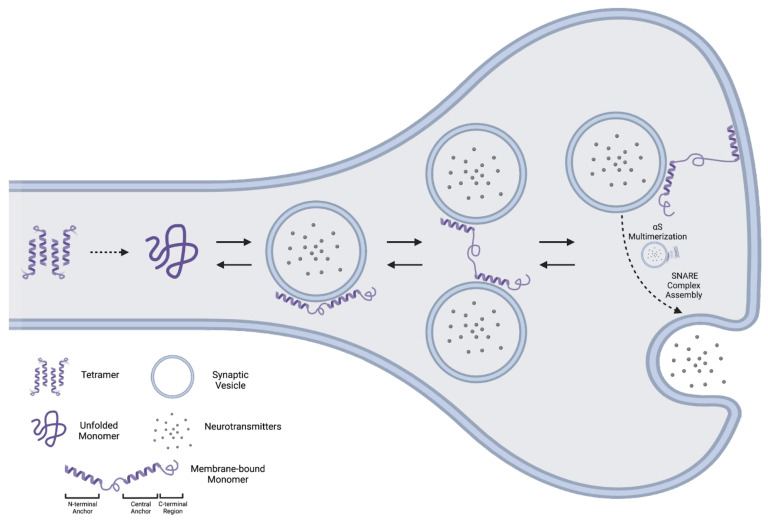
**Physiologic conformations of αS.** Steady-state populations of cytosolic and membrane-associated αS exist in the cell. Cytosolic αS exists as an unfolded monomer, or, controversially, a folded tetramer (far left), while membrane-bound αS adopts a compact folded structure. Membrane association is stabilized by two α-helical anchors on the N-terminal and central regions of the protein. αS’ anchors may be associated with either the same vesicle (middle left), two different vesicles, promoting vesicle clustering (middle right), or a vesicle and the plasma membrane, tethering vesicles to membrane sites (far right, top). At the presynaptic membrane, αS promotes SNARE complex assembly, the catalytic machinery for synaptic vesicle fusion and neurotransmitter release (far right, bottom). The self-assembly of αS into higher-order “multimers” may be required for this latter process.

**Figure 2 biomolecules-12-00994-f002:**
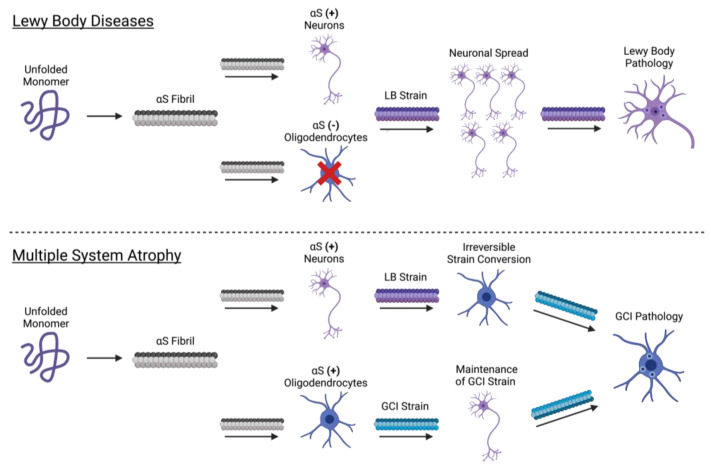
**αS strains.** Pathologically, αS aggregation results in the formation of amyloid fibrils with prion-like properties. Conformationally distinct populations of αS fibrils exist, termed “strains”. In vivo, cell-type-specific factors determine the αS strain(s) formed. The propagation of different αS strains leads to distinct pathologic patterns of spread and, therefore, the distinct clinical manifestations of the synucleinopathies. In Lewy body diseases (i.e., LBD and PD) (top panel), the infection of a neuron with αS fibrils results in templated aggregation of αS monomer and, due to neuron-specific cell factors, the formation of the LB strain of αS. The cell-to-cell spread of the LB strain results in the characteristic lipid-rich cytoplasmic inclusions in neurons termed Lewy bodies. In MSA (bottom panel), oligodendrocytes abnormally express or extracellularly acquire αS. The infection of these αS-expressing oligodendrocytes results in the formation of the GCI-strain of αS. Likewise, the infection of neurons results in the formation of the LB-strain of αS. However, oligodendrocytes irreversibly convert the LB-strain to the GCI-strain, whereas neurons are incapable of the reverse process (GCI-strain to LB-strain). Thus, the GCI-strain becomes the predominant αS strain, the cell-to-cell spread of which results in the characteristic glial cytoplasmic inclusions of MSA.

**Figure 3 biomolecules-12-00994-f003:**
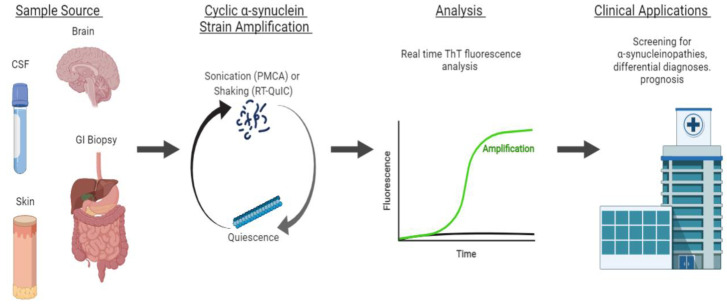
**Strain amplification assay for αS.** Sample sources include the brain, cerebrospinal fluid (CSF), skin, and gastrointestinal (GI) biopsies. From the sample, αS fibrils are amplified in cycles of sonication (PMCA) or shaking (RT-QuIC) to make polymerization points available for the misfolding of wild-type proteins. The sonication and shaking periods are followed by a period of quiescence, where the fibrils aggregate and polymerize. This process is cyclically repeated to amplify misfolded proteins to detectable levels that are analyzed by the real-time monitoring of ThT fluorescence. This technique can be applied clinically for the early screening, differential diagnoses, and prognosis of α-synucleinopathies.

## Data Availability

Not applicable.

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
