# Peer review of "α-Synuclein Conformational Plasticity: Physiologic States, Pathologic Strains, and Biotechnological Applications"

_biomolecules, 2022, doi:10.3390/biom12070994_

Round 1

Reviewer 2 Report

This is a nicely written review discussing the conformational nature of α-synuclein, and its implications for α-synuclein function in health and disease. The discussion of the controversy surrounding tetramers is clear and important. The section on amplification assays, especially strain specificity and the potential to use it to distinguish between MSA and PD, was particularly interesting. Overall, a very interesting and enjoyable to read manuscript. I have only a few suggestions to make the story more comprehensive.

-No mention is really made that pathologic α-synuclein differs from physiological in more than being monomer/tetramer vs higher order oligomer. While a complete review of phosphor/ubiquitin/nitration modifications in pathological aS is not needed, it should at least be touched upon, particularly as to possible effects on the aggregation properties.

-The discussion of amplification assays in different tissues would be improved if some mention of the distribution of aS expression throughout the body were made. A number of groups are already trying to use PMCA/RT-QuIC in other tissues as well, so even a brief mention of the possibility of using other samples (e.g., blood, which has abundant aS) would be relevant.

-Several studies have considered aS in exosomes/extracellular vesicles. This has implications both for the conformation of aS normally, and for its pathological spread, and is worth some discussion as well.

Round 2

Reviewer 1 Report

Thanks for taking into consideration all the proposed changes.